# Reliable Post hoc Explanations:
# Modeling Uncertainty in Explainability

**Dylan Slack**
UC Irvine
dslack@uci.edu

**Sophie Hilgard**
Harvard University
ash798@g.harvard.edu

**Sameer Singh**
UC Irvine
sameer@uci.edu

**Himabindu Lakkaraju**
Harvard University
hlakkaraju@hbs.edu

## Abstract

As black box explanations are increasingly being employed to establish model credibility in high stakes settings, it is important to ensure that these explanations are accurate and reliable. However, prior work demonstrates that explanations generated by state-of-the-art techniques are inconsistent, unstable, and provide very little insight into their correctness and reliability. In addition, these methods are also computationally inefficient, and require significant hyper-parameter tuning. In this paper, we address the aforementioned challenges by developing a novel Bayesian framework for generating local explanations along with their associated uncertainty. We instantiate this framework to obtain Bayesian versions of LIME and KernelSHAP which output credible intervals for the feature importances, capturing the associated uncertainty. The resulting explanations not only enable us to make concrete inferences about their quality (e.g., there is a 95% chance that the feature importance lies within the given range), but are also highly consistent and stable. We carry out a detailed theoretical analysis that leverages the aforementioned uncertainty to estimate how many perturbations to sample, and how to sample for faster convergence. This work makes the first attempt at addressing several critical issues with popular explanation methods in one shot, thereby generating consistent, stable, and reliable explanations with guarantees in a computationally efficient manner. Experimental evaluation with multiple real world datasets and user studies demonstrate that the efficacy of the proposed framework.[1]

## 1 Introduction

As machine learning (ML) models get increasingly deployed in domains such as healthcare and criminal justice, it is important to ensure that decision makers have a clear understanding of the behavior of these models. However, ML models that achieve state-of-the-art accuracy are typically complex *black boxes* that are hard to understand. As a consequence, there has been a surge in post hoc techniques for explaining black box models [1–10]. Most popular among these techniques are local explanation methods which explain complex black box models by constructing interpretable local approximations (e.g., LIME [2], SHAP [4], MAPLE [11], Anchors [1]). Due to their generality, these methods are being leveraged to explain a number of classifiers including deep neural networks and ensemble models in a variety of domains such as law, medicine, and finance [12, 13].

Existing local explanation methods, however, suffer from several drawbacks. Explanations generated using these methods may be unstable [14–18], i.e., negligibly small perturbations to an instance can result in substantially different explanations. These methods are also inconsistent [19] i.e., multiple runs on the same input instance with the same parameter settings may result in vastly different explanations. There are also no reliable metrics to ascertain the quality of the explanations

---

[1]Project Page: https://dylanslacks.website/reliable/index.html

35th Conference on Neural Information Processing Systems (NeurIPS 2021).

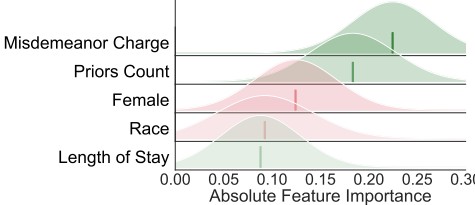 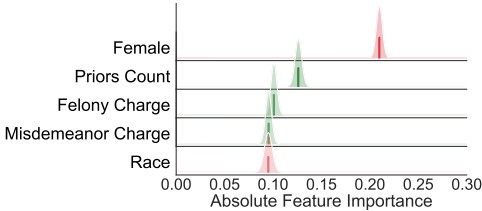

(a) Explanation computed with 100 perturbations     (b) Explanation with 2000 perturbations

Figure 1: **Example explanations** on for an instance from the COMPAS dataset, where vertical lines indicate the feature importance by LIME (red is negative effect, green is positive) and the shaded region visualizes the uncertainty estimated by BayesLIME. While LIME produces very different and contradictory feature importance for different number of perturbations (1a and 1b), BayesLIME provides more context. The overlapping uncertainty intervals in the explanation computed with 100 perturbations (1a) indicate that it is unclear which feature is the most important. However, the tighter uncertainty intervals in the explanation computed with 2K perturbations (1b) clearly indicates that `Female` is the most important.

output by these methods. Commonly used metrics such as explanation fidelity rely heavily on the implementation details of the explanation method (e.g., the perturbation function used in LIME) and do not provide a true picture of the explanation quality [20]. Furthermore, there exists little to no guidance on determining the values of certain hyperparameters that are critical to the quality of the resulting local explanations (e.g., number of perturbations in case of LIME). Local explanation methods are also computationally inefficient i.e., they typically require a large number of black box model queries to construct local approximations [21]. This can be prohibitively slow especially in case of complex neural models.

In this paper, we identify that modeling uncertainty in black box explanations is the key to addressing all the aforementioned challenges. To this end, we propose a novel Bayesian framework for generating local explanations along with their associated uncertainty. We instantiate this framework to obtain Bayesian versions of LIME and KernelSHAP, namely BayesLIME and BayesSHAP, that not only output point-wise estimates of feature importance but also their associated uncertainty in the form of credible intervals (See Figure 1). We derive closed form expressions for the posteriors of the explanations thereby eliminating the need for any additional computational complexity. The credible intervals produced by our framework not only allow us to make concrete inferences about the quality of the resulting explanations but also produce explanations that satisfy user specified levels of uncertainty (e.g., an end user may request for explanations that satisfy a certain 95% confidence level). In addition, the resulting explanations are also highly consistent and stable. *To the best of our knowledge, this work makes the first attempt at addressing several critical challenges in popular explanation methods in one-shots, thereby generating consistent, stable, and reliable explanations with guarantees in a computationally efficient manner.*

We carry out theoretical analysis that leverages the measures of uncertainty (credible intervals) produced by our framework to estimate the values of critical hyperparameters. More specifically, we derive a closed form expression for the number of perturbations required to generate explanations that satisfy desired levels of confidence. We also propose a novel sampling technique called *focused sampling* that leverages uncertainty to determine how to sample perturbations for faster convergence, thereby enabling our framework to generate explanations in a computationally efficient manner.

We evaluate the efficacy of the proposed framework on a variety of datasets including COMPAS, German Credit, ImageNet, and MNIST. Our results demonstrate that the explanations output by our framework are not only highly reliable, but also very consistent and stable (53% more stable than LIME/SHAP on an average). Our experimental results also confirm that we can accurately estimate the number of perturbations needed to generate explanations with a desired level of uncertainty, and that our uncertainty sampling technique speeds up the process of generating explanations by up to a factor of 2 relative to random sampling of perturbations. Lastly, we carry out a user study with 31 human subjects to evaluate the quality of the explanations generated by our framework, demonstrating that our explanations accurately capture the importance of the most influential features.

## 2 Notation & Background

Here we introduce notation and discuss two relevant prior approaches, LIME and KernelSHAP.

**Notation** Let $f : \mathbb{R}^d \to [0, 1]$ denote a black box classifier that takes a data point $x$ with $d$ features, and returns the *probability* that $x$ belongs to a certain class. Our goal is to explain individual predictions of $f$. Let $\phi \in \mathbb{R}^d$ denote the explanation in terms of feature importances for the prediction $f(x)$, i.e. coefficients $\phi$ are treated as the feature *contributions* to the black box prediction. Note that $\phi$ captures the coefficients of a linear model. Let $\mathcal{Z}$ be a set of $N$ randomly sampled instances (perturbations) around $x$. The proximity between $x$ and any $z \in \mathcal{Z}$ is given by $\pi_x(z) \in \mathbb{R}$. We denote the vector of these distances over the $N$ perturbations in $\mathcal{Z}$ as $\Pi_x(\mathcal{Z}) \in \mathbb{R}^N$. Let $Y \in [0, 1]$ be the vector of the black box predictions $f(z)$ corresponding to each of the $N$ instances in $\mathcal{Z}$.

**LIME** [2] and **KernelSHAP** [4] are popular *model-agnostic local explanation* approaches that explain predictions of a classifier $f$ by learning a linear model $\phi$ locally around each prediction (i.e. $y \sim \phi^T z$). The objective function for both LIME and KernelSHAP constructs an explanation that approximates the behavior of the black box accurately in the vicinity (neighborhood) of $x$.

$$\arg\min_{\phi} \sum_{z \in \mathcal{Z}} [f(z) - \phi^T z]^2 \pi_x(z). \tag{1}$$

The above objective function has the following closed form solution:

$$\hat{\phi} = (\mathcal{Z}^T \text{diag}(\Pi_x(\mathcal{Z}))\mathcal{Z} + \mathbb{I})^{-1} (\mathcal{Z}^T \text{diag}(\Pi_x(\mathcal{Z}))Y) \tag{2}$$

The main difference between LIME and KernelSHAP lies in how $\pi_x(z)$ is chosen. In LIME, it is chosen heuristically: $\pi_x(z)$ is computed as the cosine or $l_2$ distance. KernelSHAP leverages game theoretic principles to compute $\pi_x(z)$, guaranteeing that explanations satisfy certain properties.

## 3 Our Framework: Bayesian Local Explanations

In this section, we introduce our Bayesian framework which is designed to capture the uncertainty associated with local explanations of black box models. First, we discuss the generative process and inference procedure for the framework. Then, we highlight how our framework can be instantiated to obtain Bayesian versions of LIME and SHAP. Lastly, we present detailed theoretical analysis for estimating the values of critical hyperparameters, and discuss how to efficiently construct highly accurate explanations with uncertainty guarantees using our framework.

### 3.1 Constructing Bayesian Local Explanations

Our goal here is to explain the behavior of a given black box model $f$ in the vicinity of an instance $x$ while also capturing the uncertainty associated with the explanation. To this end, we propose a Bayesian framework for constructing local linear model based explanations and capturing their associated uncertainty. We model the black box prediction of each perturbation $z$ as a linear combination of the corresponding feature values ($\phi^T z$) plus an error term ($\epsilon$) as shown in Eqn (4). While the weights of the linear combination $\phi$ capture the feature importances and thereby constitute our explanation, $\epsilon$ captures the error that arises due to the mismatch between our explanation $\phi$ and the local decision surface of the black box model $f$. Our complete generative process is shown below:

$$y|z, \phi, \epsilon \sim \phi^T z + \epsilon \qquad \epsilon \sim \mathcal{N}(0, \frac{\sigma^2}{\pi_x(z)}) \tag{3}$$

$$\phi|\sigma^2 \sim \mathcal{N}(0, \sigma^2 \mathbb{I}) \qquad \sigma^2 \sim \text{Inv-}\chi^2(n_0, \sigma_0^2). \tag{4}$$

The error term is modeled as a Gaussian whose variance relies on the proximity function $\pi_x(z)$ i.e., $\epsilon \sim \mathcal{N}(0, \frac{\sigma^2}{\pi_x(z)})$. This proximity function ensures that perturbations closer to the data point $x$ are modeled accurately, while allowing more room for error in case of perturbations that are farther away. $\pi_x(z)$ can be computed using cosine or $l_2$ distance or other game theoretic principles similar to that of LIME and KernelSHAP (see Section 2). The conjugate priors on $\phi$ and $\sigma^2$ are shown in Eqn (4). Note that, the distributions on error $\epsilon$ and feature importance $\phi$ both consider the parameter $\sigma^2$. The

fact that the prior on the feature importances considers $\sigma^2$ has an intuitive interpretation: if we have prior knowledge that the error of the explanation is small, we expect to be more confident about the feature importances. Similarly, if we have prior knowledge the error is large, we expect to be less confident about the feature importances.

Thus, our generative process corresponds to the Bayesian version of the weighted least squares formulation of LIME and KernelSHAP outlined in Eqn. (1), with additional terms to model uncertainty. As in Eqns. (4), the process captures two sources of uncertainty in local explanations: 1) ***feature importance uncertainty***: the uncertainty associated with the feature importances $\phi$, and (2) ***error uncertainty***: the uncertainty associated with the error term $\epsilon$ which captures how well our explanation $\phi$ models the local decision surface of the underlying black box.

**Inference** Our inference process involves estimating the values of two key parameters: $\phi$ and $\sigma^2$. By doing so, we can compute the local explanation as well as the uncertainties associated with feature importances and the error term. Posterior distributions on $\phi$ and $\sigma^2$ are normal and scaled Inv-$\chi^2$, respectively, due to the corresponding conjugate priors [22]:

$$\sigma^2 | \mathcal{Z}, Y \sim \text{Scaled-Inv-}\chi^2 \left( n_0 + N, \frac{n_0 \sigma_0^2 + N s^2}{n_0 + N} \right)$$

$$\phi | \sigma^2, \mathcal{Z}, Y \sim \text{Normal}(\hat{\phi}, V_\phi \sigma^2) \tag{5}$$

Further, $\hat{\phi}$, $V_\phi$, and $s^2$ can be directly computed:

$$\hat{\phi} = V_\phi (\mathcal{Z}^T \text{diag}(\Pi_x(\mathcal{Z})) Y)$$

$$V_\phi = \left( \mathcal{Z}^T \text{diag}(\Pi_x(\mathcal{Z})) \mathcal{Z} + \mathbb{I} \right)^{-1} \tag{6}$$

$$s^2 = \frac{1}{N} \left[ (Y - \mathcal{Z}\hat{\phi})^T \text{diag}(\Pi_x(\mathcal{Z}))(Y - \mathcal{Z}\hat{\phi}) + \hat{\phi}^T \hat{\phi} \right] \tag{7}$$

Details of the complete inference procedure including derivations of Eqns. (5-7) are provided in the Appendix A. Note that our estimate of the posterior mean feature importances $\hat{\phi}$ (Eqn. (6)) is the same as that of the feature importances computed in case of LIME and KernelSHAP (Eqn. (2)).

**Remark 3.1.** *If we use the same proximity function $\pi_x(z)$ in our framework as in LIME or KernelSHAP, the posterior mean of the feature importance $\hat{\phi}$ output by our framework (Eq (6)) will be equivalent to the feature importances output by LIME or KernelSHAP, respectively.*

**Feature Importance Uncertainty** To obtain the local feature importances and their associated uncertainty, we first compute the posterior mean of the local feature importances $\hat{\phi}$ using the closed form expression in Eqn. (7). We then estimate the credible interval (measure of uncertainty) around the mean feature importances by repeatedly sampling from the posterior distribution of $\phi$ (Eq (5)).

**Error Uncertainty** The error term $\epsilon$ can serve as a proxy for explanation quality because it captures the mismatch between the constructed explanation and the local decision surface of the underlying black box. We first calculate the marginal posterior distribution of $\epsilon$ by leveraging Eqn (4) and integrating out $\sigma^2$. This results in a three parameter Student's t distribution (derivation in appendix A):

$$\epsilon | \mathcal{Z}, Y \sim t_{(\mathcal{V} = n_0 + N)}(0, \frac{n_0 \sigma_0^2 + N s^2}{n_0 + N}). \tag{8}$$

We then evaluate the probability density function (PDF) of the above posterior at 0, i.e., $P(\epsilon = 0)$ by substituting the value of $s^2$ computed using Eqn. (7) into the Student's t distribution above (Eqn. (8)). The resulting expression gives us the probability density that the explanation output by our framework perfectly captures the local decision surface underlying the black box. This operation is performed in constant time, adding minimal overhead to non-Bayesian LIME and SHAP. We illustrate how these computed intervals capture the variance in the explanations in Figure 9.

**Proposition 3.2.** *As the number of perturbations around $x$ goes to $\infty$ i.e., $N \to \infty$: (1) the estimate of $\phi$ converges to the true feature importance scores, and its uncertainty to $0$. (2) uncertainty of the error term $\epsilon$ converges to the bias of the local linear model $\phi$. [Details in Appendix B]*

**BayesLIME and BayesSHAP** Our framework can be instantiated to obtain the Bayesian version of LIME by setting the proximity function to $\pi_x(z) = \exp(-D(x, z)^2/\sigma^2)$ where $D$ is a distance metric

(e.g. cosine or $l_2$ distance), and $n_0$ and $\sigma_0^2$ to small values ($10^{-6}$) so that the prior is uninformative. We compute feature importance uncertainty and error uncertainty for LIME's feature importances.

Our framework can also be instantiated to obtain the Bayesian version of KernelSHAP by setting uninformative prior on $\sigma^2$ and $\pi_x(z) = \frac{d-1}{(d \text{ choose } |z|)|z|(d-|z|)}$ where $|z|$ denotes the number of the variables in the variable combination represented by the data point $z$ i.e., the number of non-zero valued features in the vector representation of $z$. Note that the original SHAP method views the problem of constructing a local linear model as estimating the Shapley values corresponding to each of the features [4]. These Shapley values represent the contribution of each of the features to the black box prediction i.e., $f(x) = \phi_0 + \sum \phi_i$. Therefore, the measures of uncertainty output by our method BayesSHAP capture the reliability of the estimated variable contributions.

To encourage BayesLIME and BayesSHAP explanations to be sparse, we can use dimensionality reduction or feature selection techniques as used by LIME and SHAP to obtain the top K features [2, 4, 23]. We can then construct our explanations using the data corresponding to these top K features.

## 3.2 Estimating the Number of Perturbations

One of the major drawbacks of approaches such as LIME and KernelSHAP is that they do not provide any guidance on how to choose the number of perturbations, a key factor in obtaining reliable explanations in an efficient manner. To address this, we leverage the uncertainty estimates output by our framework to compute *perturbations-to-go* ($G$), an estimate of how many *more* perturbations are required to obtain explanations that satisfy a desired level of certainty. This estimate thus *predicts* the computational cost of generating an explanation with a desired level of certainty and can help determine whether it is even worthwhile to do so. The user specifies the confidence level of the credible interval (denoted as $\alpha$) and the *maximum* width of the credible interval ($W$), e.g. "width of 95% credible interval should be less than 0.1" corresponds to $\alpha = 0.95$ and $W = 0.1$. To estimate $G$ for the local explanation of a data point $x$, we first generate $S$ perturbations around $x$ (where $S$ is small and chosen by the user) and fit a local linear model using our method[2]. This provides initial estimates of various parameters shown in Eqns (5)-(7) which can then be used to compute $G$.

**Theorem 3.3.** *Given $S$ seed perturbations, the number of additional perturbations required ($G$) to achieve a credible interval width $W$ of feature importance for a data point $x$ at user-specified confidence level $\alpha$ can be computed as:*

$$G(W, \alpha, x) = \frac{4s_S^2}{\bar{\pi}_S \times \left[\frac{W}{\Phi^{-1}(\alpha)}\right]^2} - S \tag{9}$$

*where $\bar{\pi}_S$ is the average proximity $\pi_x(z)$ for the $S$ perturbations, $s_S^2$ is the empirical sum of squared errors (SSE) between the black box and local linear model predictions, weighted by $\pi_x(z)$, as in (7), and $\Phi^{-1}(\alpha)$ is the two-tailed inverse normal CDF at confidence level $\alpha$.*

*Proof (Sketch).* To estimate $G$, we first relate $W$ and $\alpha$ to $\text{Var}(\phi_i)$, the marginal variance of the feature importance[3] for any feature $i$, obtained by integrating out $\sigma^2$. Because Student's t can be approximated by a Normal distribution for large degrees of freedom (here, $S$ should be large enough), we use the inverse normal CDF to calculate credible interval width at level $\alpha$. We compute $V_\phi$ from (6) using $\mathcal{Z}$, treating its entries as Bernoulli distributed with probability $0.5$. Due to the covariance structure of this sampling procedure, the resulting variance estimate after $N$ samples is the sample SSE $s_S^2$ scaled by $\approx \frac{4}{\bar{\pi}_S N}$ (derivation in appendix B). If we assume SSE scales linearly with $S$, we can take this to be a reasonable estimate of $s_N^2$ at any $N$. We can then estimate $G$ as

$$\left[\frac{W}{\Phi^{-1}(\alpha)}\right]^2 = \text{Var}(\phi_i) = \frac{4s_S^2}{\bar{\pi}_S \times (G+S)} \implies G = \frac{4s_S^2}{\bar{\pi}_S \times \left[\frac{W}{\Phi^{-1}(\alpha)}\right]^2} - S. \tag{10}$$

$\square$

---

[2]We assume a simplified feature space where features are present or absent according to Bernoulli(.5). As in Ribeiro et al. [2], these *interpretable* features are flexible and can encode what is important to the end user.

[3]Since the error depends primarily on the number of perturbations, $\text{Var}(\phi_i)$ is similar across features.

### 3.3 Focused Sampling of Perturbations

Perturbations-to-go ($G$) provides us with an estimate of how many samples are required to achieve reliable explanations. However, if $G$ is large, querying the black-box model for its predictions on a large number of perturbations can be computationally expensive for larger models [24, 25]. To reduce this cost, we develop an alternative sampling procedure called *focused sampling* which leverages uncertainty estimates to query the black box in a more targeted fashion (instead of querying randomly), thereby reducing the computational cost associated with generating reliable explanations. Inspired by active learning [26], focused sampling strategically prioritizes perturbations whose predictions the explanation is most uncertain about, when querying the black box. This enables the focused sampling procedure to query the black box only for the predictions of the most informative perturbations and thereby learn an accurate explanation with far fewer queries to the black box.

To determine how uncertain our explanation $\phi$ is about the black box label for any given instance $z$, we first compute the posterior predictive distribution for $z$ (derivation in Appendix A), given as $\hat{y}(z)|\mathcal{Z}, Y \sim t_{(\mathcal{V}=N)}(\hat{\phi}^T z, (z^T V_\phi z + 1)s^2)$. The variance of this three parameter student's t distribution is,

$$\text{var}(\hat{y}(z)) = ((z^T V_\phi z + 1)s^2)(N/(N-2)) \tag{11}$$

We refer to this variance as the *predictive variance* $\text{var}(\hat{y}(z))$, and it captures how uncertain our explanation $\phi$ is about the black box prediction.

The focus sampling procedure first fits the explanation with an initial $S$ perturbations (where $S$ is a small number). We then iterate the following procedure until the desired explanation certainty level is reached. We draw a batch of $A$ candidate perturbations, compute their predictive variance with the Bayesian explanation, and induce a distribution over the perturbations by running softmax on the variances with tempurature parameter $\tau$. We draw a batch of $B$ perturbations from this distribution and query the black box model for their labels. Finally, we refit the Bayesian explanation on all the labeled perturbations collected so far. We provide pseudocode for the uncertainty sampling procedure in Algorithm 1.

---

**Algorithm 1** Focused sampling for local explanations

---

**Require:** Model $f$, Data instance $x$, Number of perturbations $N$, Number of seed perturbations $S$, Batch size $B$, Pool size $A$, tempurature $\tau$

1: **function** FOCUSED SAMPLE
2:     Initialize $\mathcal{Z}$ with $S$ seed perturbations.
3:     Fit $\hat{\phi}$ on $\mathcal{Z}$                                                      ▷ Using Eqn (6)
4:     **for** $i \leftarrow 1$ to $N - S$ in increments of $B$ **do**
5:         $\mathcal{Q} \leftarrow$ Generate $A$ candidate perturbations
6:         Compute $\text{var}(\hat{y}(z))$ on $\mathcal{Q}$                            ▷ Using Eqn (11)
7:         Define $\mathcal{Q}_{\text{dist}}$ as $\propto \exp(\text{var}(\hat{y}(z))/\tau)$
8:         $\mathcal{Q}_{\text{new}} \leftarrow$ Draw $B$ samples from $\mathcal{Q}_{\text{dist}}$
9:         $\mathcal{Z} \leftarrow \mathcal{Z} \cup \mathcal{Q}_{\text{new}}$; Fit $\hat{\phi}$ on $\mathcal{Z}$                     ▷ Using Eqn (6)
10:     **end for**
11:     **return** $\hat{\phi}$
12: **end function**

---

## 4 Experiments

We evaluate the proposed framework by first analyzing the quality of our uncertainty estimates i.e., feature importance uncertainty and error uncertainty. We also assess our estimates of required perturbations ($G$), and evaluate the computational efficiency of focused sampling. Last, we describe a user study with 31 subjects to assess the informativeness of the explanations output by our framework.

**Setup** We experiment with a variety of real world datasets spanning multiple applications (e.g., criminal justice, credit scoring) as well as modalities (e.g., structured data, images). Our first structured dataset is **COMPAS** [27], containing criminal history, jail and prison time, and demographic attributes of 6172 defendants, with class labels that represent whether each defendant was rearrested

|  | **BayesLIME** | **BayesSHAP** |  | **BayesLIME** | **BayesSHAP** |
|---|---|---|---|---|---|
| TABULAR DATASETS |  |  | MNIST |  |  |
| COMPAS | 95.5 | 87.9 | Digit 1 | 95.8 | 98.4 |
| German Credit | 96.9 | 89.6 | Digit 2 | 95.8 | 97.4 |
| IMAGENET |  |  | Digit 3 | 95.2 | 96.3 |
| Corn | 94.6 | 91.8 | Digit 4 | 97.2 | 90.1 |
| Broccoli | 91.4 | 89.2 | Digit 5 | 95.2 | 95.6 |
| French Bulldog | 94.8 | 89.9 | Digit 6 | 96.7 | 96.8 |
| Scuba Diver | 92.4 | 94.6 | Digit 7 | 95.7 | 95.3 |

Table 1: **Evaluating Credible Intervals.** We report the % of time the 95% credible intervals with 100 perturbations include their true values (estimated on $10,000$ perturbations). Closer to $95.0$ is better. Both BayesLIME and BayesSHAP are well calibrated.

within 2 years of release. The second structured dataset is the **German Credit** dataset from the UCI repository [28] containing financial and demographic information (including account information, credit history, employment, gender) for 1000 loan applications, each labeled as a "good" or "bad" customer. We create 80/20 train/test splits for these two datasets, and train a random forest classifier (sklearn implementation with 100 estimators) as *black box* models for each (test accuracy of $82.8\%$ and $72.5\%$, respectively). We also include popular image datasets–MNIST and Imagenet. For the **MNIST** [29] handwritten digits dataset, we train a 2-layer CNN to predict the digits (test accuracy of $99.2\%$). For **Imagenet** [30], we use the off-the-shelf VGG16 model [31] as the black box. We select a sample of 100 images of the following classes French Bulldog, Scuba Diver, Corn, and Broccoli to use in the experiments. For generating explanations, we use standard implementations of the baselines LIME and KernelSHAP with default settings [2, 4]. For images, we construct super pixels as described in [2] and use them as features (number of super pixels is fixed to 20 per image). For our framework, the desired level of certainty is expressed as the width of the 95% credible interval.

**Quality of Uncertainty Estimates** A critical component of our explanations is the feature importance uncertainty. To evaluate the correctness of these estimates, we compute how often *true* feature importances lie within the $95\%$ credible intervals estimated by BayesLIME and BayesSHAP. Note, that by *true* feature importance, we refer to the best fit linear model output using either the LIME or SHAP kernels. We evaluate the quality of our credible interval estimates by running our methods with 100 perturbations to estimate feature importances and taking the corresponding 95% credible intervals for each test instance. We compute what fraction of the true feature importances fall within our 95% credible intervals. Note, because there are no methods to provide uncertainty estimates for LIME and SHAP, we do not provide further baselines. Since we do not have access to the true feature importances of the complex black box models, following Prop 3.2, we use feature importances computed using a large value of $N$ ($N = 10,000$), and treat the resulting estimates as ground truth.

Results for BayesLIME in Table 1 indicate that the true feature importances are close to ideal and indicate the estimates are well calibrated. While the estimates by BayesSHAP are somewhat less calibrated (true feature importances fall within our estimated 95% credible intervals about 89.2 to 98.4% of the time), they still are quite close to ideal. All in all, these results confirm that the credible intervals learned by our methods are well calibrated and therefore highly reliable in capturing the uncertainty of the feature importances. Lastly, though we set our priors to be uninformative in general, we also investigate how sensitive our uncertainty estimates are to hyperparameter choices in Figure 5 in the Appendix. We find that the explanation uncertainty becomes uncalibrated with strong priors. However, our explanations seem to be robust to hyperparameter choices in general.

**Correctness of Estimated Number of Perturbations** We assess whether our estimate of *perturbations-to-go* ($G$; Section 3.2) is an accurate estimate of the *additional* number of perturbations needed to reach a desired level of feature importance certainty. We carry out this experiment on MNIST data for the digit "4" (additional datasets explored in Appendix C) and use $S = 200$ as the initial number of perturbations to obtain a preliminary explanation and its associated uncertainty estimates. We then leverage these estimates to compute $G$ for 6 different certainty levels. First, we observe significant differences in $G$ estimates across instances (details in appendix C) i.e. number of perturbations needed to obtain a particular level of certainty varied significantly across instances–

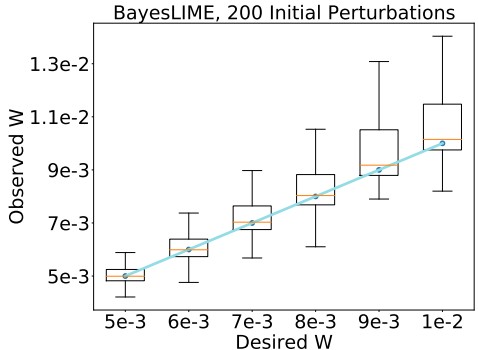
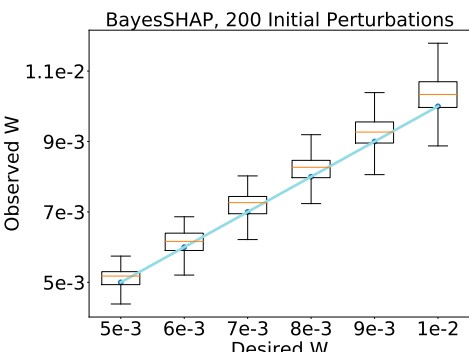

Figure 2: **Perturbations-to-go** ($G$). We generate explanation with $G$ perturbations, where $G$ is computed using the *desired* credible interval width (x-axis), and compare desired levels to the *observed* credible interval width (y-axis) (blue line indicates ideal calibration). Results are averaged over 100 MNIST images of the digit "4" We see that $G$ provides a good approximation of the additional perturbations needed.

ranging from 200-5,000 for the lowest level of certainty to 200-20,000 for higher levels of certainty. Next, for each image and certainty level, we run our method for the estimated number of perturbations ($G$) to determine if the observed estimates of uncertainty (observed credible interval width $W$) match the desired levels of uncertainty (desired credible interval width $W$). Results in Figure 2 show that the observed and desired levels of certainty are well calibrated, demonstrating that $G$ estimates are reliable approximations of the additional number of perturbations needed.

**Efficiency of Focused Sampling** *Focused sampling* uses the *predictive variance* to strategically choose perturbations that will reduce uncertainty in order to be labeled by the black box (section 3.3). Here, we will evaluate the efficiency of the focused sampling procedure. First, we assess whether focused sampling converges (as measured by error uncertainty ($P(\epsilon = 0)$)) more efficiently than random sampling. To this end, we experiment with BayesLIME on Imagenet data for the "French bulldog" class to carry out this analysis. This setting replicates scenarios where LIME is applied to a computationally expensive black box model, making it highly desirable to limit the number of perturbations to reduce total running time. We run each sampling strategy for 2,000 perturbations and plot the number of model queries versus error uncertainty. During focused sampling, we set the batch size $B$ to 50. The results in Figure 3 show that focused sampling results in faster convergence to reliable and high quality explanations; focused sampling stabilizes within a couple hundred model queries while random sampling takes over 1,000. Note, as the inefficiency of querying the black box model increases, the advantages of focused sampling decreasing total running time of the explanations will only become more pronounced. These results clearly demonstrate that focused sampling can significantly speed up the process of generating high quality local explanations. Additionally, in Appendix C, we also check if focused sampling causes any bias (due to sampling based on uncertainty estimates) that results in convergence to a different/wrong explanation, however our results clearly indicate that this is not the case.

**Stability of BayesLIME & BayesSHAP** Recall that LIME & SHAP are not stable: small changes to instances can produce substantially different explanations. We consider whether BayesLIME & BayesSHAP produce more stable explanations than their LIME & SHAP counterparts. To perform this analysis, we use the local Lipschitz metric for explanation stability [18]:

$$\hat{L}(x_i) = \underset{x_j \in N_\epsilon(x_i)}{\text{argmax}} \frac{||\phi_i - \phi_j||_2}{||x_i - x_j||_2} \tag{12}$$

where $x_i$ refers to an instance, $N_\epsilon(x_i)$ is the $\epsilon$-ball centered at $x_i$, and $\phi_i$ and $\phi_j$ are the explanation parameters for $x_i$ and $x_j$. Lower values indicate more stable explanations. We follow the setup outline by Alvarez-Melis and Jaakkola [18] and compute the local Lipschitz values, comparing both LIME & BayesLIME and SHAP & BayesSHAP across Compas, German Credit, MNIST digit "4", and Imagenet "French Bulldog." We perform the comparison using the default number of perturbations in

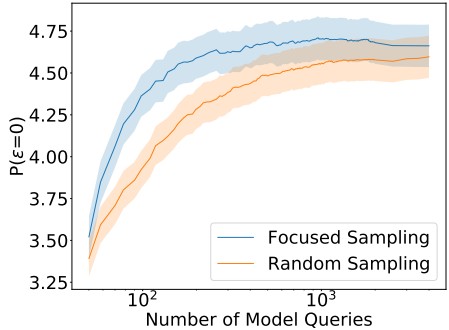

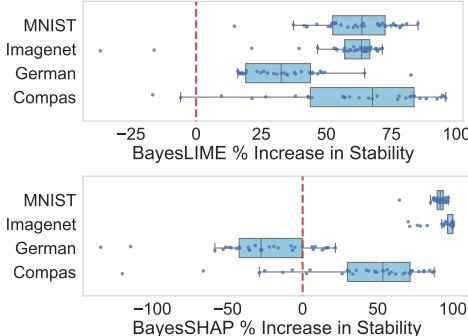

Figure 3: **Efficiency of focused sampling** for 100 Imagenet "French bulldog" images, with random sampling as a baseline. We provide mean and standard error. We assess the efficiency of focused sampling by comparing *error uncertainty* over model queries and show quicker convergence than random sampling.

Figure 4: **Assessing the % increase in stability** of BayesLIME and BayesSHAP over LIME and SHAP respectively. Our Bayesian methods are significant more stable ($\rho < $ 1e-2 according to Wilcoxon signed-rank test) except for BayesSHAP on German Credit, where there is not a significant difference between the methods ($\rho > 0.05$).

both LIME & SHAP, and use this same number in the respective Bayesian variants and set the batch size $B$ to half this value. We use focused sampling for BayesLIME and BayesSHAP, and report the % increase in stability of these approaches over LIME and SHAP for 40 test points. The results given in Figure 4 show a clear improvement (on average 53%) in stability in all cases except German Credit for BayesSHAP. Further, we run a Wilcoxon signed-rank test and find our results are statistically significant in all cases ($\rho < $ 1e-2) except for BayesSHAP for German Credit, where there is not a significant difference between the methods ($\rho > 0.05$). These results demonstrate BayesLIME and BayesSHAP are more stable than previous methods.

**User Study** We perform a user study with 31 subjects to compare BayesLIME and LIME explanations on MNIST. We evaluate the following: are explanations with low levels of uncertainty (i.e., most confident explanations) more meaningful to humans? To answer this question, we follow prior work and mask the most important features selected by BayesLIME and LIME [32, 4]. We ask users to guess the digit of the masked images. The better the explanation, the more difficult it should be for the users to get it right. Further, the choice to mask the important features is motivated by its success in prior work. We randomly select 15 correctly predicted test images, generate explanations by sweeping over a range of perturbation amounts $[10^{.5}, ..., 10^{3.5}]$ incremented by 0.5. We choose the *top* explanation for each image based on either fidelity (for LIME) or $P(\epsilon = 0)$ (for BayesLIME). We sent the user study out to students and researchers with background in computer science. A screen shot of the task is shown in Figure 7 in the Appendix. We find that the explanations output by our methods focus on more informative parts of the image, since hiding them makes it difficult for humans to guess the digit. Users had an error rate of 25.7% for LIME, while it was 30.7% for BayesLIME, both with standard error 0.003 ($\rho = 0.028$ through a one-tailed two sample t-test). This result indicates that our method BayesLIME and the associated measure of explanation uncertainty result in more high quality and reliable explanations compared to LIME and its associated fidelity metric.

## 5   Related Work

**Interpretability Methods** A variety of interpretability methods have been proposed. Some methods that are inherently interpretable include additive models [33, 34], decision lists and sets [35, 36], and instance-based explanations [37]. However, black-box models are often more flexible, accurate, and easier to use; thus, there has been a lot of interest in constructing post hoc explanations[38]. These include LIME [2] and SHAP [4, 39], which are among the most popular due to their broad applicability and code availability, but saliency maps [5–8], permutation feature importance [40], and partial dependency plots [41] also follow this paradigm. Other approaches to post hoc explanations focus on rule-based models [1, 3], counterfactuals [42, 43], and influence functions [9].

**Vulnerabilities of Post hoc Explanations** Recent work has shed light on the downsides of post hoc explanation techniques. These methods are often highly sensitive to small changes in inputs [14], are susceptible to manipulation [15, 16, 44, 45], and are not faithful to the underlying black boxes [46]. Perturbation-based explanation methods such as LIME and SHAP are subject to additional criticisms: results vary between runs of the algorithms [18–20, 47, 21], and hyperparameters used to select the perturbations can greatly influence the resulting explanation [20]. Prior work has attempted to tackle the problem of instability in perturbation-based explanations by averaging over several explanations [48, 19], however, this is computationally expensive. Other works related to creating more trustworthy explanations include development of sanity checks for explainers [49, 17, 50]. These techniques represent an important step towards improved usability, given experimental evidence that humans are often too eager to accept inaccurate machine explanations [51–54]. Recent works theoretically analyze the sources of non-robustness in black box explanations [55–57].

**Logical and Formal Reasoning** Additional related works have considered explaining classifiers through identifying a subset of features that are "sufficient" to explain a prediction [58–62]. Though these methods offer strong guarantees surrounding which features ensure a prediction is achieved, they are not model agnostic. Further, they do not define feature importances associated with the local explanations nor consider ways to improve locally weighted explanations, such as LIME and SHAP.

**Bayesian Methods in Explainable ML** Few recent works have adopted Bayesian formulations to explain black box models [63–65]. Guo et al. [63] introduce a Bayesian non-parametric approach to fit a *global* surrogate model. Their formulation seeks to fit a mixture of generalizable explanations across instances. Zhao et al. [64] study whether incorporating informative priors improves the stability of the resulting explanations. However, neither of these works focus on modeling the uncertainty of local explanations. Further, these approaches also do not tackle the critical problems of estimating key hyperparameters or improving efficiency of computing explanations.

## 6 Conclusion

We developed a Bayesian framework for generating local explanations along with their associated uncertainty. We instantiated this framework to obtain Bayesian versions of LIME and SHAP that output pointwise estimates of feature importances as well as their associated credible intervals. These intervals enabled us to infer the quality of the explanations and output explanations that satisfied user specified levels of uncertainty. We carried out theoretical analysis that leverages these uncertainty measures (credible intervals) to estimate the values of critical hyperparameters (e.g., the number of perturbations). We also proposed a novel sampling technique called focused sampling that leverages uncertainty estimates to determine how to sample perturbations for faster convergence.

While the Bayesian framework addresses several critical challenges (i.e., consistency, stability, modeling uncertainty) associated with LIME and SHAP, there are still certain aspects where it would exhibit the same shortcomings as LIME and SHAP [4, 66]. For instance, if the local decision surface of a given black box classifier is highly non-linear, our framework, which relies on local linear approximations, may not be able to capture this non-linear decision surface accurately. In addition, if the perturbation sampling procedures used in LIME and SHAP are used in BayesLIME and BayesSHAP, they will likely vulnerable to the attacks proposed by Slack et al. [15]. In the future, it would be interesting to extend our framework to produce global explanations with uncertainty guarantees and explore how uncertainty quantification can help calibrate user trust in model explanations.

## 7 Acknowledgments

We would like to thank the anonymous reviewers for their insightful feedback. This work is supported in part by the NSF awards #IIS-2008461, #IIS-2008956, and #IIS-2040989, and research awards from the Harvard Data Science Institute, Amazon, Bayer, Google, and the HPI Research Center in Machine Learning and Data Science at UC Irvine. The views expressed are those of the authors and do not reflect the official policy or position of the funding agencies.

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
