# A Derivations

**Model Derivation**  We write the joint posterior as

$$\phi, \sigma^2 | Y, \mathcal{Z} \propto \rho(Y|X, \beta, \sigma^2)\rho(\beta|\sigma^2)\rho(\sigma^2) \tag{13}$$

$$\propto (\sigma^2)^{-N/2}\exp(-\frac{1}{2\sigma^2}(Y - \mathcal{Z}\phi)^T \text{diag}(\Pi_x(\mathcal{Z}))\cdot$$

$$(Y - \mathcal{Z}\phi))(\sigma^2)^{-1}\exp(-\frac{1}{2\sigma^2}\phi^T\phi)(\sigma^2)^{-(1+\frac{n_0}{2})}\exp\left[\frac{-n_0\sigma_0^2}{2\sigma^2}\right] \tag{14}$$

Letting $\hat{\phi} = (\mathcal{Z}^T\text{diag}(\Pi_x(\mathcal{Z}))\mathcal{Z} + I)^{-1}\mathcal{Z}^T\text{diag}(\Pi_x(\mathcal{Z}))Y$, we group terms in the exponentials according to $\phi$. The intermediate steps can be found in [67]. Supressing dependence on $Y$ and $\mathcal{Z}$, we can write down the conditional posterior of $\phi$ as

$$\phi|\sigma^2 \propto \exp(\frac{1}{2}\sigma^{-2}[\phi - \hat{\phi}]^T(\mathcal{Z}^T\text{diag}(\Pi_x(\mathcal{Z}))\mathcal{Z} + I)[\phi - \hat{\phi}]) \tag{15}$$

So, we can see that our estimates for the mean and variance of $\rho(\phi|\sigma^2, Y, \mathcal{Z})$ are $\hat{\phi}$ and $\sigma^2(\mathcal{Z}^T\text{diag}(\Pi_x(\mathcal{Z}))\mathcal{Z} + I)^{-1}$. Next, we derive the conditional posterior for $\sigma^2$. We identify the form of the scaled inverse-$\chi^2$ distribution in the joint posterior as in [22] and write

$$\sigma^2|\hat{\phi} \sim \text{Inv-}\chi^2(N + n_0, \frac{n_0\sigma_0^2 + Ns^2}{n_0 + N}) \tag{16}$$

where $s^2$ is defined as in equation 7.

**Derivation of equation 8**  We establish the identity [22]:

$$\sigma^2 \sim \text{Inv-}\chi^2(a, b) \text{ and } z|\sigma^2 \sim \mathcal{N}(\mu, \lambda\sigma^2)$$
$$\iff z \sim t_{(\mathcal{V}=a)}(\mu, \lambda b) \tag{17}$$

We have, $\epsilon \sim \mathcal{N}(0, \sigma^2)$, $\sigma^2 \sim \text{Inv-}\chi^2(N + n_0, \frac{n_0\sigma_0^2 + Ns^2}{n_0 + N})$. Then, it's the case that $\epsilon \sim t_{(\mathcal{V}=N+n_0)}(0, \frac{n_0\sigma_0^2 + Ns^2}{n_0 + N})$.

**Derivation of Posterior Predictive**  Note, this derivation takes the priors to be set as in BayesLIME or BayesSHAP, namely, with values close to zero. We apply the identity from equation 17 to derive this posterior. We have $\hat{y} \sim \hat{\phi}^T z + \epsilon$ for some $z$. Thus, $\hat{y} \sim \mathcal{N}(\hat{\phi}^T z, z^T V_\phi z\sigma^2) + \mathcal{N}(0, \sigma^2)$, where $\sigma^2 \sim \text{Inv-}\chi^2(N, s^2)$. So, we have $\hat{y} \sim t_{(\mathcal{V}=N)}(\hat{\phi}^T z, (z^T V_\phi z + 1)s^2)$.

# B Proof of Theorems

In these derivations, the perturbation matrices $\mathcal{Z}$ have elements $\mathcal{Z}_{ij} \in \{0, 1\}$ where each $\mathcal{Z}_{ij} \sim$ Bernoulli(0.5). Note, in these proofs, we take take the priors to be set as in BayesLIME and BayesSHAP, i.e., they have hyperparameter values close to 0.

## B.1 Proof of Theorem 3.3

Note that we use $N$ to denote the *total* perturbations while $S$ denotes the perturbations collected *so far*. We use three assumptions stated as follows. First, $\frac{\bar{\pi}N}{2}$ is sufficiently large such at $\frac{\bar{\pi}N}{2} + 1$ is equivalent to $\frac{\bar{\pi}N}{2}$. Second, $N$ is sufficiently large such that $N + 1$ is equivalent to $N$ and $\frac{N}{N-2}$ is equivalent to 1. Third, the product of $\mathcal{Z}^T\text{diag}(\Pi_x(\mathcal{Z}))\mathcal{Z}$ within $V_\phi$ can be taken at its expected value. First, we state the marginal distribution over feature importance $\phi_i$ where $i$ is an arbitrary feature importance $i \in d$. This given as

$$\phi_i|\mathcal{Z}, Y \sim t_{\mathcal{V}=N}(\hat{\phi}_i, V_{\phi_{ii}}s^2) \tag{18}$$

where $V_\phi = (\mathcal{Z}^T\text{diag}(\Pi_x(\mathcal{Z}))\mathcal{Z} + I)^{-1}$. Recall each $\mathcal{Z}_{ij}$ is given $\sim$ Bern(.5) we use the third assumption to write $V_\phi$ is $\frac{\bar{\pi}N}{2} + 1$ for the on diagonal elements and $\frac{\bar{\pi}N}{4}$ for the off diagonal elements.

We can see this is the case considering that each element in $\mathcal{Z}$ is a Bern$(.5)$ draw. We drop the $1's$ due to the first assumption.

Let $k = \frac{\bar{\pi}N}{2}$. It follows directly from Sherman Morrison that the $i$-th and $j$-th entries of $V_\phi$ are given as

$$(V_\phi)_{ij} = \begin{cases} \frac{2}{k} - \frac{2}{k(N+1)} & i = j \\ -\frac{2}{k(N+1)} & i \neq j \end{cases} \quad (V_\phi)_{ii} = \frac{4}{\bar{\pi}(N+1)} \tag{19}$$

We see that the diagonals are the same. Thus, we take the $PTG$ estimate in terms of a single marginal $\phi_i$. Substituting in the $s^2$ estimate $s_S^2$ and using the second assumption, we write the variance of marginal $\phi_i$ as

$$\text{Var}(\phi_i) = \frac{4s_S^2}{\bar{\pi}(N+1)}\frac{N}{N-2} \tag{20}$$

$$= \frac{4s_S^2}{\bar{\pi} \times N} = \frac{4s_S^2}{\bar{\pi} \times \text{Var}(\phi_i)} \tag{21}$$

Because feature importance uncertainty is in the form of a credible interval, we use the normal approximation of $\text{Var}(\phi_i)$ and write

$$N = \frac{4s_S^2}{\bar{\pi} \times \left[\frac{W}{\Phi^{-1}(\alpha)}\right]^2} \tag{22}$$

where $W$ is the desired width, $\alpha$ is the desired confidence level, and $\Phi^{-1}(\alpha)$ is the two-tailed inverse normal CDF. Finally, we subtract the initial $S$ samples. $\qquad \square$

### B.2   Proposition 3.2

Before providing a proof for proposition 3.2, we note to readers that the claims are related to well known results in bayesian inference (e.g. similar results are proved in [68]). We provide the proofs here to lend formal clarity to the properties of our explanations.

**Convergence of Var$(\phi)$**   Recall the posterior distribution of $\phi$ given in equation 5. In equation 19, we see the on and off-diagonal elements of $V_\phi$ are given as $\frac{4}{\bar{\pi}(N+1)}$ and $-\frac{4}{\bar{\pi}N(N+1)}$ respectively (here replacing $S$ with $N$ to stay consistent with equation 5). Because we have $N \to \infty$, these values define $V_\phi$ due to the law of large numbers. Thus, as $N \to \infty$, $V_\phi$ goes to the null matrix and so does the uncertainty over $\phi$.

**Consistency of $\hat{\phi}$**   Recall the mean of $\phi$, denoted $\hat{\phi}$ given in equation 6. To establish consistency, we must show that $\hat{\phi}$ converges in probability to the true $\hat{\phi}$ as $N \to \infty$. To avoid confusing true $\hat{\phi}$ with the distribution over $\phi$, we denote the true $\hat{\phi}$ as $\phi^*$. Thus, we must show $\hat{\phi} \to_p \phi^*$ as $N \to \infty$. We write

$$\hat{\phi} = (\mathcal{Z}^T \text{diag}(\Pi_x(\mathcal{Z}))\mathcal{Z} + I)^{-1}\mathcal{Z}^T \text{diag}(\Pi_x(\mathcal{Z}))Y \tag{23}$$

$$= (\mathcal{Z}^T \text{diag}(\Pi_x(\mathcal{Z}))\mathcal{Z} + I)^{-1}\mathcal{Z}^T \text{diag}(\Pi_x(\mathcal{Z}))(\mathcal{Z}\phi^* + \epsilon) \tag{24}$$

Considering mean of $\epsilon$ is 0 and using law of large numbers,

$$= (\mathcal{Z}^T \text{diag}(\Pi_x(\mathcal{Z}))\mathcal{Z} + I)^{-1}\mathcal{Z}^T \text{diag}(\Pi_x(\mathcal{Z}))\mathcal{Z}\phi^* = \phi^* \tag{25}$$

**Convergence of Var$(\epsilon)$**   Assume we have $N \to \infty$ so $\hat{\phi}$ converges to $\phi^*$. The uncertainty over the error term is given as the variance of the distribution in equation 8. The variance of this generalized student's t distribution is given as converges to $s^2$ for large $N$. Recalling its definition, $s^2$ reduces to the local error of the model as $N \to \infty$. which is equivalent to the squared bias of the local model.

## C   Detailed Results

In this appendix, we provide extended experimental results.

### C.1 Explanation Uncertainty Hyperparameter Sensitivity

In the main paper, we assume the priors are set to be uninformative. Though this is the advised configuration for BayesLIME and BayesSHAP because prior information about the local surface is not likely available, we assess the calibration sensitivity of BayesLIME to different choices in the hyperparameters. In figure 5, we perform a grid search over the uncertainty hyperparameters $n_0$ and $\sigma_0^2$ for the MNIST digit "4" class. We find the explanation uncertainty is robust to the choice of hyperparameters.

| $n_0$ \ $\sigma_0^2$ | $1e-5$ | $1e-1$ | $1$ | $10$ | $100$ |
|---|---|---|---|---|---|
| $1e-5$ | 95.7 | 95.7 | 96.4 | 96.6 | 96.6 |
| $1e-1$ | 96.6 | 96.6 | 96.9 | 98.9 | 100.0 |
| $1$ | 96.5 | 96.9 | 98.6 | 100.0 | 100.0 |
| $10$ | 94.2 | 98.2 | 100.0 | 100.0 | 100.0 |
| $100$ | 72.2 | 99.0 | 100.0 | 100.0 | 100.0 |

Figure 5: BayesLIME calibration sensitivity to the choice of hyperparameters. Closer to $95.0$ is better. These results indicate BayesLIME calibration is not very sensitive to choices in the hyperparameter values.

### C.2 PTG Estimate Results

In the main paper, we provided PTG results for BayesLIME on MNIST. In this appendix, we show the number of perturbations estimated by PTG and additional PTG results on the Imagenet "French bulldog" class.

**Number of Perturbations Estimated by PTG**    In section 4, we assessed if $G$ produces good estimates of the number of additional samples needed to reach the desired level of feature importance certainty. In figure 6, we show the desired level of certainty (desired width of credible interval $W$) versus the actual $G$ estimate (i.e. the estimated number of perturbations) for figure 2 in the main paper. We see the estimated number of perturbations is highly variable depending on desired $W$.

**Further PTG Estimate Results**    We provide results for the PTG estimate on Imagenet in Figure 8. We limit the range of uncertainty values compared to MNIST because the Imagenet data is more complex and consequently the required number of perturbations becomes very high. These results further indicate the effectiveness of the PTG estimate.

### C.3 User Study

**Participate Consent**  We sent out an email to students and researchers with a background in computer science inviting them to take our user study. At the beginning of the user study, we stated that no

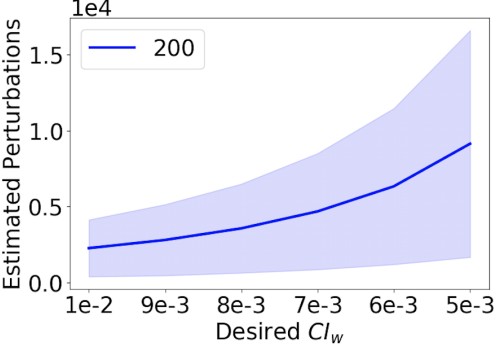

Figure 6: Desired $CI_w$ versus the actual number of perturbations estimated by $PTG$ in figure 2 of the main paper. We plot mean and standard deviation of $G$.

personal information would be asked during the study, their answers would be used in a research project, and whether they consented to take the study.

**Image of interface** We give an example screen shot from the user study in Figure 7.

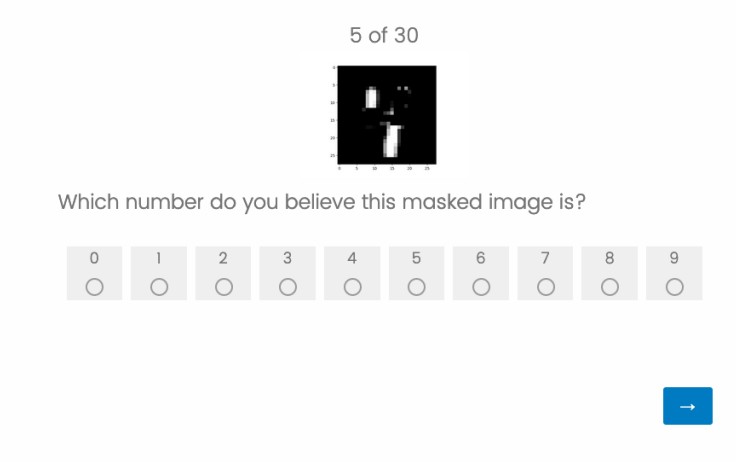

Figure 7: Screen shot from user study (correct answer 4).

**BayesLIME Toy Example** To show how our bayesian methods capture the uncertainty of local explanations, we provide an illustrative example in Figure 9. Rerunning LIME explanations on a toy decision surface (blue lines in the figure), we see LIME has high variance and produces many different explanations. This behavior is particularly sever in the nonlinear surfaces. With a single explanation, BayesLIME captures the uncertainty associated with generating local explanations (black lines in the figure).

## C.4  Focused Sampling Results

In this appendix, we provide additional focused sampling results. We include a comparison of focused sampling to random sampling in terms of wall clock time. We also provide results demonstrating the focused sampling procedure is not biased.

**Wall Clock Time of Focused Sampling** In figure 10, we plot wall clock time versus $P(\epsilon = 0)$. This experiment is analogous to figure 3 in the main paper, but here we use time instead of number of

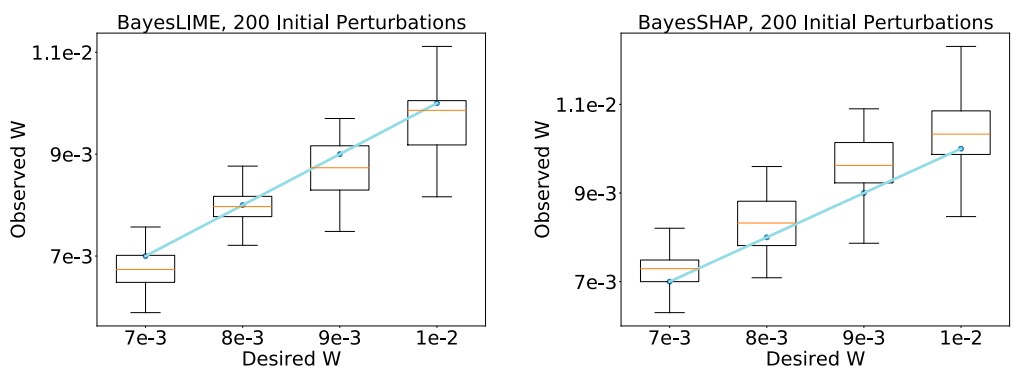

Figure 8: Imagenet PTG results for BayesLIME & BayesSHAP. The blue line indicates ideal calibration. These results indicate the PTG estimate is well calibrated for BayesLIME and BayesSHAP on Imagenet, demonstrating the efficacy of the estimate.

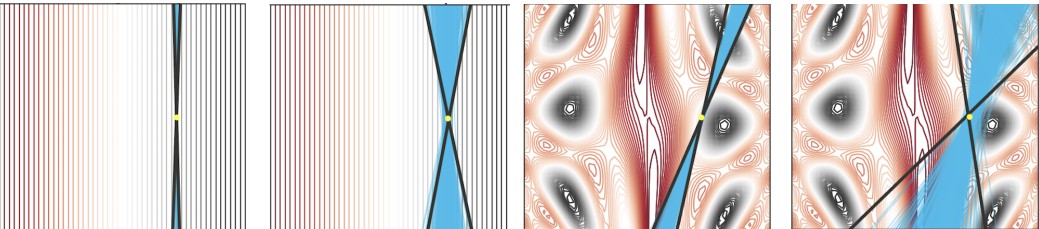

| (a) linear, many samples | (b) linear, fewer samples | (c) nonlinear, many samples | (d) nonlinear, fewer samples |

Figure 9: Rerunning LIME local explanations 1000 times and BayesLIME *once* for linear and non-linear toy surfaces using few (25) and many (250) perturbations. The linear surface is given as $p(y) \propto x_1$ and the non linear surface is defined as $p(y) \propto \sin(x_1/2) * 10 + \cos(10 + (x_1 * x_2)/2) * \cos(x_1)$. We plot each run of LIME in blue and the BayesLIME 95% credible region of the feature importance $\phi$ in black. We see that LIME variance is higher with fewer samples and a less linear surface. BayesLIME captures the relative difficulty of explaining each surface through the width the credible region. For instance, BayesLIME is most uncertain in the nonlinear, few samples case because this surface is the most difficult to explain.

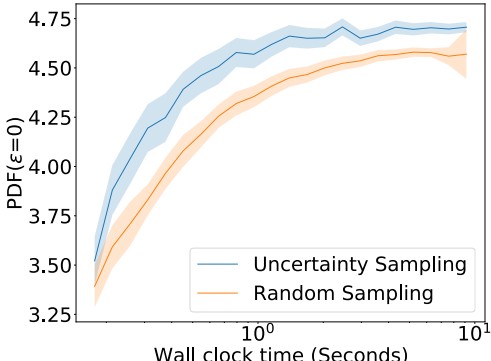

Figure 10: Wall clock time needed to converge to a high quality explanation by BayesLIME (analogous to figure 3 in the paper). We use both random sampling and focused sampling over 100 Imagenet images. We provide the mean and standard error for binned estimates of these values. This result demonstrates that focused sampling leads to improved convergence over random sampling in terms of wall clock time.

model queries on the x-axis. We see that uncertainty sampling is more time efficient than random sampling for BayesLIME.

**Bias of Focused Sampling** In the main text, we saw that focused sampling converges faster than random sampling. However, it is possible that focused sampling introduces bias into the process due to sampling based on uncertainty estimates, leading to convergence to a different/wrong explanation. To assess whether this occurs in practice, we evaluate the convergence of both focused sampling and random sampling to the "true" explanation on Imagenet (computed with the number of perturbations $N = 10,000$ using random sampling). To measure convergence, we compare the $L_1$ distance of the explanation with the ground truth explanation. The result provided in Figure 11 demonstrates that focused sampling converges to the ground truth explanation with significantly fewer model queries than random sampling. Focused sampling reaches a $L_1$ distance of 0.1 at 300 queries while it takes upwards of 450 queries for random sampling, indicating improved query efficiency of $30 - 40\%$. Lastly, as the number of model queries increases ($\sim$1000), we observe an $L_1$ distance of around 0.06 which is extremely small and the explanations are practically the same as the ground truth. Overall, these results show that focused sampling does not suffer from biases in practice and further demonstrate that focused sampling can lead to significant speedups.

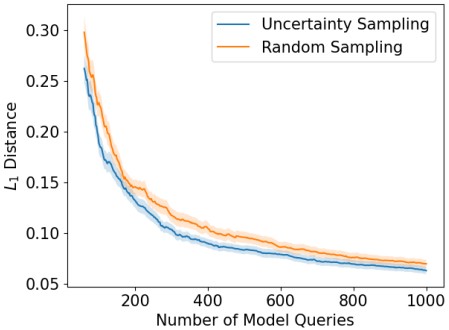

Figure 11: Convergence of both focused sampling and random sampling to the ground truth explanation. We see that uncertainty sampling converges more quickly to the ground truth than random sampling, demonstrating that there is minimal bias in the focused sampling procedure and focused sampling converges more efficiently.

### C.5 Benchmarking

We also benchmark the efficiency of BayesLIME and BayesSHAP against Guo et al. [63], a related Bayesian explanation method that uses a Bayesian non parametric mixture regression and MCMC for parameter inference. Fixing their mixture regression to a single component results in a similar model to ours and thus is a useful point of comparison. To explain a single instance on ImageNet using VGG16, their approach takes 139.2 seconds, while BayesLIME and BayesSHAP take 20.3 seconds and 21.1 seconds respectively, under the same conditions, demonstrating that the closed form solution is very efficient.

## D Explaining a Ground Truth Function

We consider a synthetic experiment in which we observe an underlying ground truth function and verify that lower values of feature importance uncertainty indicate higher proximity between the feature importance estimates and the underlying ground truth function. To this end, we constructed a piecewise linear function of two variables, where each quadrant in the x,y-plane corresponds to a different linear model. We consider the regression coefficients of the quadrant as the ground truth explanation. The piecewise function is given as:

$$
\begin{aligned}
f(x, y) = {}& 0.3x + 0.2y \text{ if } x > 0, y > 0 \\
& 0.2x - 0.1y \text{ if } x > 0, y \le 0 \\
& -x - 0.05y \text{ if } x \le 0, y \le 0 \\
& -.8x + 0.2y \text{ if } x \le 0, y > 0
\end{aligned}
\tag{26}
$$

We plot the $\ell_1$ distance between the BayesLIME feature importance mean and ground truth explanation versus the maximum credible interval width of the BayesLIME explanation. The results given in Figure 12 indicate that tighter credible intervals lead to explanations that are closer to the ground truth, demonstrating that the feature importance uncertainties are meaningful in regards to a ground truth function.

## E Compute Used

In this work, we ran all experiments on a single NVIDIA 2080TI & a single NVIDIA Titan RTX GPU.

## F Dataset licenses

German Credit is in the public domain, COMPAS uses the MIT license, MNIST uses the Creative Commons Attribution-Share Alike 3.0 license, and Imagenet does not hold copyright of images.

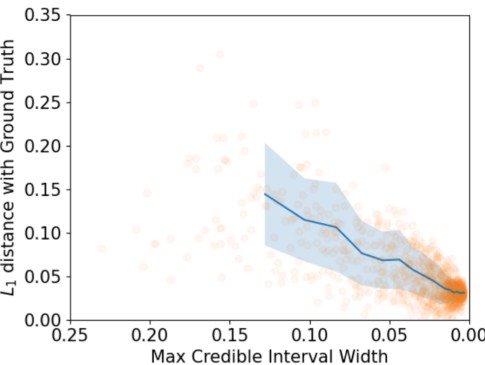

Figure 12: Assessing whether tighter credible intervals lead to convergence with ground truth, on an example where the ground truth feature importances are known. Here, we plot The $\ell_1$ distance between the feature importances menas for BayesLIME and ground truth explanation versus the maximum credible interval width across the explanation.