# OpenReview forum: "Reliable Post hoc Explanations: Modeling Uncertainty in Explainability"
_NeurIPS.cc/2021/Conference — NeurIPS 2021 Poster_

### Official Review · Reviewer_w3MC · 2021-07-12

**Rating:** 7
**Confidence:** 3

**Summary:**

This paper addresses uncertainty in local explanation methods, i.e., methods that use a local surrogate model which is used to generate explanations of black-box models. Methods such as LIME and KernelSHAP are well-known examples. The main idea is to a Bayesian framework for constructing local explanations and capturing its uncertainty. Key is a distribution over an error which arises due to the mismatch between the explanation and the local decision surface of the black box model. The resulting method allow for computing uncertainty intervals for features, but also allows for estimating the number of perturbations needed to generate better explanations, and more focused sampling of additional samples around the local decision area.


**Limitations And Societal Impact:**

Yes, the conclusions hint at the possibility that the credible intervals are incorrect, which may lead to overreliance of faulty explanations. It would not be hard to see that this could be damaging in some domains. Nonetheless, the work is fairly theoretical, so a further consideration is not necessary.

**Main Review:**

In my opinion, this paper contains a significant contributions: despite their obvious advantages, methods such as LIME suffer
from a number of downsides which are mostly related to instability and non-robustness of the explanations. The experiments in this
paper show that this criticism can mostly be overcome using a Bayesian approach to these ideas.

The novelty of the paper is reasonable: while it builds upon ideas of LIME and KernelSHAP, the development and derivation of
the whole framework is novel as far as I can see.

The evaluation is convincing. I appreciated the evaluation of the credible intervals, showing that for BayesSHAP and certainly for BayesLIME the 95% credible interval is close to the true value. The results show that it is a significant improvement over the baseline local explanation methods, both in more theoretical terms and in a small-scale test with humans.

The paper is quite easy to read. It contains some technical details, but even if one were to skip those, it is not hard to get the main ideas of the approach.

**Time Spent Reviewing:**

2

---

> ### Author Response · Authors · 2021-08-11
> **Response to Reviewer v3MC**
>
> We appreciate the positive feedback concerning the significance of our contribution, the novelty of our work, the quality of our evaluation, and writing. Thank you!

---

### Official Review · Reviewer_TaGg · 2021-07-13

**Rating:** 3
**Confidence:** 4

**Summary:**

This paper proposes an approach to computing local explanations for machine learning (ML) predictions that are argued to be more reliable and accurate than the explanations provided by the prior approaches of LIME, Anchors, MAPLE, and SHAP. The developed solution attributes responsibility for the prediction by estimating feature importance as well as studying the credible intervals. The approach is based on a "Bayesian framework" and is instantiated to devise improved versions of LIME and KernelSHAP referred to in the paper as BayesLIME and BayesSHAP, respectively. Similarly to the other explainers, the approach is based on extensive sampling in the vicinity of the instance being explained but, in contrast, it can estimate the remaining amount of sampling needed to fulfil a specific level of certainty. The reported experimental results demonstrate that the explanations returned by the new approach are more accurate and more stable than those of LIME and SHAP. Additionally, a user study is performed indicating that the approach is superior to the competition in this regard.

**Limitations And Societal Impact:**

See the main discussion above.

**Main Review:**

This is yet another line of work to propose doing extensive sampling in a neighbourhood of a given instance to compute local explanations of the prediction made for that instance. The approach aims at addressing the known issues of the existing explainers of the same nature, namely the explanations being inaccurate and/or imprecise and/or unstable. Here, I should say the paper is somewhat hard to follow and it does not make a convincing case, at least to me. I believe the approach is not clearly explained and the rationale used is not detailed either - instead, the authors throw a large number of statistical formulas and variable/constant symbols one after another with the gaps filled by some short connecting phrases. (The appendix looks similar.) I am not sure to what extent such a description can be deemed sufficient for a top conference in AI/ML like NeurIPS.

The major problem I have with this paper is that it ignores a vast range of work in the area of logic and formal reasoning applied to computing provably correct (that is *truly reliable*) explanations, including "sufficient reasons" and "probabilistic sufficient reasons", and also assessing the correctness of explanation using formal reasoning. (I believe these must be hard to ignore and they include works published at flagship AI conferences, e.g. IJCAI/AAAI/ICML/NeurIPS.) This problem is especially troublesome in light of the reliability claims made here. For instance, it is known that in many cases, explanations computed by LIME/Anchors/SHAP are plain wrong, which can be easily established by applying formal reasoning to the explanations. (Precision of Anchors has been also assessed in prior works and it turned out to be unsatisfactory in a number of cases.) I fail to see any discussion on whether the proposed approach is free of this issue. And I believe such a discussion is necessary due to the claims made.

Experimental results on the accuracy of explanations rely on using 10000 samples - I am not sure why it can be deemed sufficient to be taken as "ground truth". Once again, unless I am missing something, one can apply probabilistic sufficient reasons (or exact/approximate model counting for formal representations of the decision function) to devise the guarantees provided by their explanations.

Also, I fail to see a discussion on whether or not the proposed approach is susceptible to out-of-distribution attacks. The idea was proposed by the same authors, and so I would expect they have thought about it.

The related work section ignores a huge body of SOTA work on learning interpretable models (but also initial works in the area defining those models).

I like the idea of estimating the amount of sampling, which is yet to be done to achieve some reasonable estimates, and it is good to see that it works somewhat well in practice.

It is unclear to me if there are any accuracy guarantees with respect to the focused sampling if done instead of random sampling.

I do not agree with the way the user study is set up. You do not hide the explanations - instead, you make the pixels black. In a similar way, you could have made them white. But I believe the proper way would be to highlight them with a third colour and make a user decide what parts of the region correspond to black or white.

**Time Spent Reviewing:**

8

---

> ### Author Response · Authors · 2021-08-10
> **Response to reviewer TaGg (1/2)**
>
> ## Response Part (1/2)
>
> We thank the reviewer for their insightful feedback. We are glad that the reviewer appreciates the idea of estimating the amount of sampling (number of perturbations) required to obtain reasonable model explanations. Below, we address specific questions and comments raised by the reviewer.
>
> ### Clarifying our contributions
>
> Despite well-documented concerns, LIME and SHAP continue to be among the most popular explanation methods and are being employed in critical real world applications [4,5,6,7,8,9]. Thus, while alternatives exist, we believe it is important to ask the critical question ““how can we improve these methods to generate more effective explanations?” In particular, we address the following two challenges:  Firstly, these methods lack consistency i.e., repeatedly running these methods with the same inputs and parameter settings produces different explanations (Figure 10 in appendix) and [17]) and there is no credible way to determine which explanation is the right one (i.e., the best fit linear model). Furthermore, different values of the number of perturbations result in different explanations. Secondly, these methods are unstable i.e., infinitesimally small changes to input instances result in drastically different explanations.
>
> In order to address the aforementioned questions, we consider two crucial aspects in this work: 1) Feature attributions output by LIME and SHAP do not reflect the uncertainty associated with the sampling process employed by these methods. This makes it difficult to determine which explanations are accurate. Instead of just outputting feature importances, capturing the sampling uncertainty associated with each explanation can be incredibly helpful in this regard. 2) One reason for the challenges discussed in the previous paragraph (consistency/stability) is that these approaches may not converge to the best fit local linear models when the number of perturbations being employed is too low. The alternative of employing a large number of perturbations is not practical because querying complex black box models (e.g., resnet, alexnet etc.) repeatedly for labels of a very large number of instances is computationally prohibitive. So, the key to addressing these challenges lies in determining the minimum number of perturbations required to generate an accurate explanation (i.e., the best fit local linear model). Our work addresses these two crucial aspects and significantly furthers the state-of-the-art w.r.t. post hoc explainability as acknowledged by all the other reviewers.
>
> ### Approach not clearly explained
>
> We tried our best to clearly explain (as acknowledged by other reviewers as well) the rationale behind our generative model (Lines 99 - 117), inference (Lines 118-125), Bayes LIME/SHAP (148-162), how to estimate perturbations (section 3.2), and our efficient sampling procedure (section 3.3). We would be very happy to clarify any further details about any aspect of our approach/paper.
>
> ### Related work on logic and formal reasoning/interpretable models
>
> Due to space constraints, we followed the precedent set by other state-of-the-art papers on post hoc explainability and only cited the most relevant references from the literature on post hoc explanations and interpretable models. In our related work section, we cited and briefly discussed several works pertaining to inherently interpretable models including initial works -- e.g., generalized additive models, decision lists/sets, case based models (line 327 onwards). Due to space constraints, we could not provide a detailed treatment of each of these methods. We will definitely expand on this discussion in the final version.
>
> We will definitely include additional references pertaining to logic and formal reasoning (e.g., [10] - [16] below) and discuss related work on provably correct explanations including "sufficient reasons" and "probabilistic sufficient reasons".  Thanks for the pointers! We would also like to note that the pointers you suggested on logic and formal reasoning recognize that methods such as LIME/SHAP generate flexible and useful classes of explanations. For instance, [10] states “a benefit of [model agnostic explanations] is that they can be used to explain any model and are generally more flexible and scalable than their alternatives”.
>
> ### Reliability claims
>
> We would like to clarify a potential misunderstanding regarding the usage of the term ‘reliability’ in our paper. We use the term ‘reliability’ to signify the fact that our framework captures the uncertainty associated with the resulting explanations. Instead of just outputting a vector of feature importances as is the case with LIME/SHAP, our framework generates more ‘reliable’ explanations by capturing the uncertainty associated with these feature importances. Capturing this uncertainty is incredibly helpful in determining which explanations are more likely to correspond to the best fit linear model.
>
> As we discussed earlier in this response (See “Clarifying our contributions”), the goal of this work is to improve the effectiveness of popular post hoc explanation methods such as LIME/SHAP. While our framework addresses several critical challenges (i.e., consistency, stability, the notion of reliability as discussed in the previous paragraph, scalability) associated with LIME/SHAP, there are still certain aspects where it would exhibit the same shortcomings as LIME/SHAP. For instance, if the local decision surface of a given black box classifier is highly non-linear, our framework which relies on local linear approximations (just like LIME/SHAP) may not be able to capture/model this nonlinear decision surface accurately. This is also a shortcoming exhibited by LIME/SHAP and every other explanation method [20, 21] that leverages linear models to approximate the local behavior of black box classifiers.
>
> ### Experimental Results and 10,000 Samples
>
> We would like to clarify that the term ‘ground truth’ in this context refers to the best possible local linear model that algorithms such as LIME/SHAP would converge to if they have access to an infinite number of samples from a given local neighborhood. Since it is impractical to obtain access to an infinite number of samples from a given local neighborhood, we leverage a very large (yet finite) number of small random perturbations of a given instance to construct the aforementioned ground truth. To this end, we experimented with different sample sizes and found that a sample size of 10,000 is large enough to converge to the best possible local linear model in case of datasets comprising 20 features or less. So, we ran LIME/SHAP with 10,000 perturbations to determine the “ground truth explanation” (i.e., the best possible local linear model).
>
> ###  Out-of-Distribution (OOD) attacks
>
> LIME and Kernel SHAP are susceptible to OOD attacks mainly due to the nature of the perturbation functions employed by these approaches. These functions randomly perturb a given data instance and are therefore likely to generate perturbations that are OOD. Our framework is also susceptible to these OOD attacks if we employ the same perturbation functions as LIME and Kernel SHAP. However, several new perturbation functions which are robust to such attacks have been proposed in recent literature [1, 2, 3]. These new functions ensure that the resulting perturbations lie on the data manifold. Both our framework and LIME/SHAP will be robust to OOD attacks if we employ these recently proposed perturbation functions. We will include these details in the final version.
>
> ###  Accuracy Guarantees
>
> While we do not establish theoretical guarantees on the accuracy of focused sampling, we empirically demonstrate that focused sampling converges to an unbiased estimate of the best possible local linear model much faster than random sampling (See “Efficiency of Focused Sampling in section 4).
>
> ###  User Study
>
> In our user study, we chose black pixels to hide important features (as determined by explanations) because the majority of the pixels in the MNIST data are black. Note that this strategy has already been employed in prior work to assess the quality of explanations. For example, in [18] (figure 4) the authors evaluate the quality of their feature importance estimates by masking pixels with high feature importance values using the background (majority) color and demonstrating how this makes the resulting images hard to discern. The same approach is adopted by [19]  (figure 5).  Thus, masking pixels using the background (majority) color is a well established approach to evaluate the quality of feature importance estimates.
>
> We thank you for your comments. We hope we addressed all your questions/concerns/comments adequately. In light of our clarifications, please consider increasing your score to accept. Please let us know if we can provide any further details and/or clarifications.

---

> > ### Author Response · Authors · 2021-08-11
> > **Response to reviewer TaGg (2/2)**
> >
> > ## Response (2/2)
> >
> > [1] Improving LIME Robustness with Smarter Locality Sampling, Sean Saito, Eugene Chua, Nicholas Capel, Rocco Hu Adv. ML Workshop 2020
> >
> > [2] Shapley explainability on the data manifold, Christopher Frye, Damien de Mijolla, Tom Begley, Laurence Cowton, Megan Stanley, Ilya Feige. ICLR 2021
> >
> > [3] Better sampling in explanation methods can prevent dieselgate-like deception, Domen Vreš, Marko Robnik Šikonja 2021
> >
> > [4] Radwa Elshawi, Mouaz H Al-Mallah, and Sherif Sakr.2019.On the interpretability of machine learning-based model for predicting hypertension. BMC medical informatics and decision making 19, 1 (2019), 146.
> >
> > [5] Mark Ibrahim, Melissa Louie, Ceena Modarres, and John Paisley. 2019. Global Explanations of Neural Networks: Mapping the Landscape of Predictions. In Proceedings of the 2019 AAAI/ACM Conference on AI, Ethics, and Society (AIES ’19). 279–287.
> >
> > [6] Leanne S Whitmore, Anthe George, and Corey M Hudson. 2016. Mapping chemical performance on molecular structures using locally interpretable explanations. arXiv preprint arXiv:1611.07443 (2016).
> >
> > [7] Umang Bhatt, Alice Xiang, Shubham Sharma, Adrian Weller, Ankur Taly, Yunhan Jia, Joydeep Ghosh, Ruchir Puri, José M. F. Moura, and Peter Eckersley. Explainable Machine Learning in Deployment. FAccT 2020.
> >
> > [8] https://docs.aws.amazon.com/sagemaker/latest/dg/clarify-model-explainability.html
> >
> > [9] https://www.fiddler.ai/explainable-ai
> >
> > [10] Eric Wang, Pasha Khosravi and Guy Van den Broeck. Probabilistic Sufficient Explanations, In Proceedings of the 30th International Joint Conference on Artificial Intelligence (IJCAI), 2021.
> >
> > [11] Andy Shih, Arthur Choi, and Adnan Dar- wiche. A symbolic approach to explaining bayesian network classifiers. In Proceedings of IJCAI, 2018.
> >
> > [12] Adnan Darwiche and Auguste Hirth. On the reasons behind decisions. In European Con- ference on Artificial Intelligence (ECAI), 2020.
> >
> > [13] Weijia Shi, Andy Shih, Adnan Darwiche, and Arthur Choi. On tractable representations of binary neural networks, 2020.
> >
> > [14] Yacine Izza, Alexey Ignatiev, and Joao Marques-Silva. On explaining decision trees, 2020.
> >
> > [15] Alexey Ignatiev, Nina Narodytska, and Joao Marques-Silva,
> > ‘Abduction-based explanations for machine learning models’, in Thirty-Third AAAI Conference on Artificial Intelligence (AAAI), pp. 1511–1519, (2019).
> >
> > [16] Alexey Ignatiev, Nina Narodytska, and Joao Marques-Silva, ‘On relating explanations and adversarial examples’, in Advances in Neural Information Processing Systems 32, 15883–15893, Curran Associates, Inc., (2019).
> >
> > [17] Muhammad Rehman Zafar and Naimul Mefraz Khan. DLIME: A deterministic local interpretable model- agnostic explanations approach for computer-aided dia- gnosis systems. In Proc. of SIGKDD Workshop on Ex- plainable AI/ML (XAI) for Accountability, Fairness and Transparency, page 6. ACM, 2019.
> >
> > [18] CXPlain: Causal Explanations for Model Interpretation under Uncertainty." Advances in Neural Information Processing Systems 32 (2019): 10220-10230.
> >
> > [19] Lundberg, Scott M., and Su-In Lee. "A unified approach to interpreting model predictions." Proceedings of the 31st international conference on neural information processing systems. 2017.
> >
> > [20] Marco Tulio Ribeiro, Sameer Singh, and Carlos Guestrin. “Why should i trust you?: Explaining the predictions of any classifier”. In: Proceedings of the 22nd ACM SIGKDD International Conference on Knowledge Discovery and Data Mining. ACM. 2016, pp. 1135–1144.
> >
> > [21] Sushant Agarwal, Shahin Jabbari, Chirag Agarwal, Sohini Upadhyay, Zhiwei Steven Wu, Himabindu Lakkaraju. Towards the Unification and Robustness of Perturbation and Gradient Based Explanations. ICML 2021.
> >
> > [22] Chen, Hugh, et al. "True to the Model or True to the Data?." arXiv preprint arXiv:2006.16234 (2020).

---

> > ### Comment · Reviewer_TaGg · 2021-09-02
> > **after rebuttal**
> >
> > First of all, I appreciate the authors' rebuttal. Thank you.
> >
> > I still believe the paper inherits all the "birthmarks" of LIME, SHAP, and Anchors and addresses the explainability problem from the same direction. The paper has a bold title on reliability and proposes nothing significant to reach true reliability. I accept that the proposed approach may address some problems typical to prior heuristic explanation approaches but the major problems remain. For instance, "why is this even deemed to be an explanation (and not statistic correlation)?" As I mentioned in my review, the authors must have discussed their work from the perspective of truly reliable explanations (if they opt to make the reliability claims), like sufficient reasons and probabilistic sufficient reasons. No, it's not enough to cite these works in passing - there must be a discussion and/or comparison. Also, I find it troubling to say that the paper cites the "most relevant" papers on "post hoc explanations and interpretable models" when it does not. To conclude, I believe the paper should not be accepted at this stage and I am going to keep my score.

---

> > > ### Author Response · Authors · 2021-09-02
> > > **Response to Follow Up Comments of Reviewer TaGg**
> > >
> > > We thank the reviewer for their follow up comments. Below, we address specific questions/concerns raised by the reviewer:
> > >
> > > **”Why is this even deemed to be an explanation (and not statistic correlation)?"**:
> > >
> > > Our usage of the term “explanation” is consistent with prior literature on post hoc explainability (e.g., LIME [2], SHAP [4], Anchors [1], MAPLE [11], etc. -- citations point to references in the main paper). Our approach builds on this prior literature and addresses several challenges (e.g., consistency, scalability, stability etc.) associated with existing post hoc explanation methods. More specifically, our approach generates explanations that capture both local feature importances and their associated uncertainty. Given this, we are following the conventional norms of explainability literature and using the term “explanation”.
> > >
> > > While we understand that there could be a broader debate about if the outputs generated by the aforementioned methods should be referred to as explanations (or if they should be called “statistical correlations”), our usage of the term “explanation” is consistent with the conventions established by prior works.
> > >
> > > **Related Work**:
> > >
> > >  We would like to reassure the reviewer that we will not only cite but also discuss and compare with the works on “sufficient reasons” [c1, c2], “probabilistic sufficient reasons” [c3, posted to arxiv on May 21, 2021] in the final version. We will also include detailed discussions about other relevant research on logic and formal reasoning (See References [10-16] in Response to reviewer TaGg (2/2)) in the final version. If there are other relevant works that we missed in our main paper and rebuttal, we request the reviewer to point out the specific papers, and we will gladly discuss them in the final version.
> > >
> > > We would also like to note that [c1] and [c2] differ from our line of work in several ways. First, these works focus on generating explanations for specific classes of models (e.g., Bayesian networks and neural networks that only accept binary 0 or 1 input). Further, these methods are not very scalable. On the other hand, our work focuses on generating model agnostic explanations for any black box classifier in a scalable fashion. In addition, our empirical evaluation focuses on real world datasets (e.g., COMPAS, Imagenet, MNIST etc.) and complex real world models (e.g., CNN, VGG 16) which cannot be trivially handled by the methods proposed in [c1, c2]. We will discuss all these aspects in detail and include empirical comparisons in the final version.
> > >
> > > Lastly, we would like to note that [c3] is a very recent paper (posted to arxiv on May 21, 2021 -- https://arxiv.org/abs/2105.10118; a week before the NeurIPS paper submission deadline). So, it would have been tricky for us to find this work, discuss it in detail, and compare with it in our submitted version. We will, however, include a detailed discussion and comparison with [c3] in the final version.
> > >
> > > **Reliability**:
> > >
> > > We clarified our usage of the term “reliability” in our rebuttal (See “Reliability Claims” in “Response to Reviewer TaGg (1/2)”). We would like to reiterate that we use the term ‘reliability’ **only** to signify the fact that our framework captures the uncertainty associated with the resulting explanations. Instead of just outputting a vector of feature importances as is the case with LIME/SHAP, our framework generates more ‘reliable’ explanations by capturing the uncertainty associated with these feature importances. Capturing this uncertainty is incredibly helpful in determining which explanations are more likely to correspond to the best fit linear model.
> > >
> > > We will also include the aforementioned clarifications about our usage of the term “reliability” in the final version of the paper.
> > >
> > > We thank the reviewer again for their time and effort in helping with the review of our paper. We sincerely hope that we addressed the questions/comments/concerns raised by the reviewer adequately.
> > >
> > > [c1] Andy Shih, Arthur Choi, and Adnan Darwiche. A symbolic approach to explaining bayesian network classifiers. In Proceedings of the International Joint Conference on Artificial Intelligence (IJCAI), 2018.
> > >
> > > [c2] Adnan Darwiche and Auguste Hirth. On the reasons behind decisions. In Proceedings of the European Conference on Artificial Intelligence (ECAI), 2020.
> > >
> > > [c3] Eric Wang, Pasha Khosravi and Guy Van den Broeck. Probabilistic Sufficient Explanations. To Appear in the Proceedings of International Joint Conference on Artificial Intelligence (IJCAI), 2021.

---

### Official Review · Reviewer_VUsD · 2021-07-15

**Rating:** 8
**Confidence:** 5

**Summary:**

The authors  set to address the current  challenges of many black box DL models by developing a novel Bayesian framework for generating local explanations along with their associated uncertainty.

The make use of a novel framework to obtain Bayesian versions of LIME and KernelSHAP which output credible intervals for the feature importances, capturing the associated uncertainty. The resulting explanations are also highly consistent and stable.
Furthermre, the authors carry out a detailed theoretical analysis that leverages the aforementioned uncertainty to estimate how many perturbations to sample, and how to sample for faster convergence.

This work makes the first attempt at addressing several critical issues with popular explanation methods in one shot, thereby generating consistent, stable, and reliable explanations with guarantees in a computationally efficient manner.

**Limitations And Societal Impact:**

Yes

**Main Review:**

I think this a very good contribution to ICLR given the topic and the quality of the submission (originality, contribution to the stare of the art, experimental evidence, etc) although the study might need to be supported in a more theoretical framework to make it worth of an oral presentation (I would recommend a poster or short presentation)

 Some of the strong points of the submission are summarized as follows:

1.	Studies in the interpretability of the results of deep learning models is a very important aspect, as well as the robustness of the obtained models in a variety of circumstances and under adversarial attacks.
2.	A sufficient introduction and motivations sections, but I would suggest introducing the state of the art at the beginning of the paper as it would help to get a better grasp of how the works builds upon previous work.
3.	The state of the art (despite the previous comment) contextualizes the subject matter in a succinct but comprehensive manner. Although there are certain aspects that could be improved, such as including a table outlining in a clearer manner the contributions of the authors in this context.
4.	The experimental design is good, showing a careful analysis to validate the proposal and several ablation studies to
5.	The foundations for the method are presented in great detail in a formalized manner and provides sufficient elements (i.e. examples) to assess the validity of the proposed approach.


**Time Spent Reviewing:**

3

---

> ### Author Response · Authors · 2021-08-10
> **Response to reviewer VUsD**
>
>
> We appreciate the strong positive feedback on the paper, including comments on the strength of the contribution, originality, and evaluation. Thank you! Below, we address specific comments raised by the reviewer:
>
> **State-of-the-art discussion and table summarizing contributions:** We thank the reviewer for this idea. While we discuss some of the key state-of-the-art approaches and highlight our contributions in the introduction section, we will move our related work section to the beginning of the writeup and better contextualize our contributions. As suggested by the reviewer, we will add a table in the final version that compares our approach to other state-of-the-art approaches and clearly highlights our contributions. Thank you so much for this suggestion.
>
> We thank the reviewer again for their positive feedback and insightful comments.

---

### Official Review · Reviewer_q9ZD · 2021-07-19

**Rating:** 8
**Confidence:** 4

**Summary:**

The paper proposes a bayesian framework that
1) generates local explanations whose uncertainty can be quantified.
2)  estimates how many perturbations are required to obtain an explanation (G) corresponding to a specified confidence level (W).
3) developed a new sample efficient sampling algorithm by taking advantage of the uncertainty estimates.

Given that LIME and SHAP both learn a linear surrogate model to approximate the predictions of the black-box around the neighborhood of a point, this could be formulated as bayesian regression problem.  The error is defined as a normal distribution whose mean is 0 and whose variance is a function of the inverse of proximity function $\pi_{x}(z)$. The variance is an inverse $\chi^{2}$ distribution and the feature importance given the variance is a normal distribution.  Following [1], they derive the posterior of the feature importance and the variance. Based on these posteriors, they are able to estimate the uncertainty of the feature importance and the uncertainty in estimating the error term.
Rather than perturbing at random, the points are sampled in the region where the surrogate model is the most uncertain. Thus, this reduces the number of perturbations required to generate an explanation at a specified certainty threshold.

This framework can be used to get Bayesian versions of LIME and SHAP by using proximity function $\pi_{x}(z)$ and the priors corresponding to each algorithm.

Their experiments are able to empirically reinforce their claims.

[1] Locally weighted bayesian regression, Andrew Moore


**Limitations And Societal Impact:**

1) Using only images from 1 class for MNIST and ImageNet is not sufficient to determine the effectiveness of the proposed Framework. Please use images from more classes. Similarly, using 40 data points for the stability experiment might not be significant enough for MNIST and ImageNet.

2) How effective is the uncertainty estimate of points that lie close to the boundary of the black-box classifier?

3) In table 1, could you provide some intuition as to why is the feature importance estimate of SHAP lower than LIME? Is it because of the nature of its perturbation function?

**Main Review:**

This is a very strong paper and is well written. Determining the number of perturbations for an explainability algorithm is always tricky, yet the quality of explanations depend on it . Being able to estimate G given W is  very useful.

The experiments section of the paper is very compelling. They have empirically proven the efficacy of their uncertainty estimates, G given W and sample efficient method. They further demonstrated that the sample efficient method is not biased.

**Time Spent Reviewing:**

12

---

> ### Author Response · Authors · 2021-08-10
> **Response to reviewer q9ZD**
>
> We thank the reviewer for their insightful comments and feedback. We appreciate the positive comments concerning the strength of the paper, the quality of the writing, and the compelling nature of the experiments. Thank you!
>
> Below, we address specific questions raised by the reviewer. To this end, we carried out multiple additional experiments as suggested by the reviewer.
>
> **Images from more classes:** In addition to the experimental results presented in our paper, we conducted more experiments with images from additional classes on Imagenet and MNIST datasets. More specifically, we added the following MNIST classes to our experimentation: [1, 2, 3, 5, 6] in addition to class 4. We also added the following Imagenet classes: [Scuba Diver, Corn, Broccoli] in addition to the french bulldog class. We repeated our calibration experiments (Table 1 in the main paper) using these additional classes, and found that the explanations output by BayesLIME and BayesSHAP are well calibrated even on these additional classes. Results highlighting the calibration of our explanations w.r.t. instances from each of the additional classes (e.g., MNIST 1, MNIST 2 etc.) as well as the entire set of instances aggregated across all the aforementioned classes (MNIST ALL, Imagenet ALL) are summarized in the table below. Note that values closer to 95 indicate that our estimates are well calibrated.
>
>
> | Dataset     | BayesLIME Calibration | BayesSHAP Calibration |
> |-------------|-----------------------|-----------------------|
> | MNIST 1     | 95.8                  | 98.4                  |
> | MNIST 2     | 95.8                  | 97.4                  |
> | MNIST 3     | 95.2                  | 96.3                  |
> | MNIST 5     | 95.2                  | 95.6                  |
> | MNIST 6     | 96.7                  | 96.8                  |
> | MNIST ALL | 96.0.                | 95.8                  |
> | Scuba Diver | 92.4                  | 94.6                  |
> | Corn        | 94.6                  | 91.8                  |
> | Broccoli    | 91.4                  | 89.2                  |
> | Imagenet ALL | 93.3           | 91.4                  |
>
> **Stability experiment:** We performed a Wilcoxon signed-rank test [1] and verified that our stability results are statistically significant even with 40 data points. In addition, we also re-ran our stability experiments with a higher number of data points (80 data points in total) and found that the our approaches result in significantly increased stability (the significance test returned p-values of $\rho < 1e-2$) compared to LIME and SHAP across all datasets except for German credit dataset (the p-value was $\rho > 0.05$). This is consistent with our prior results using 40 data points.
>
> **Uncertainty estimate of points that lie close to the boundary:** As suggested by the reviewer, we analyzed the effectiveness of the uncertainty estimates of points that lie close to the decision boundary of the black-box classifier. To this end, we performed our calibration experiments (Table 1 in the main paper) on points close to the boundary of the classifier using class 4 of MNIST and french bulldog class of Imagenet. To determine points close to the boundary, we take 10\% of the test set with the highest predicted entropy, i.e., $\sum_c^C - p(x)_c * \textrm{log} p(x)_c$, where c is the class and $p(x)_c$ denotes the predicted probability of the class c. We found that our uncertainty estimates are just as effective for the points close to the decision boundary. In case of MNIST, the percentage of the time that our 95% credible intervals include the true feature importance values is $96.1$ and $97.7$ for BayesLIME and BayesSHAP respectively (Note that values closer to $95$ indicate that our estimates are well calibrated). In the case of Imagenet, these values turn out to be $96.3$ and $88.1$ for BayesLIME and BayesSHAP respectively. Overall, these results indicate that our uncertainty estimates are quite well calibrated even for points close to the decision boundary.
>
> **BayesSHAP feature importance estimate:** Our intuition for the BayesSHAP calibration results is in line with that of the reviewer’s. The weighting function used by BayesSHAP tends to produce extreme values i.e., data points tend to have either extremely high or extremely low weights ( ~ $0$). We believe this contributes to the estimates from BayesSHAP having slightly worse calibration than those of BayesLIME.
>
> We thank the reviewer again for their comments, suggestions, and positive feedback on the paper.
>
> [1] https://en.wikipedia.org/wiki/Wilcoxon_signed-rank_test

---

> > ### Comment · Reviewer_q9ZD · 2021-08-19
> > **Thank you for addressing my concerns**
> >
> > The authors have addressed my concerns. I am increasing my score

---

> > > ### Author Response · Authors · 2021-08-19
> > > **Thank you!**
> > >
> > > Thank you so much for the positive feedback to our response and for increasing your score to _clear accept_! We really appreciate it.

---

### Decision · Program_Chairs · 2021-09-28

**Decision:**

Accept (Poster)

**Comment:**

There are three extremely favorable reviews and one very strong dissenting review.

The dissenting opinion (after good discussion) is: LIME, SHAP , Anchors etc..  - why are these appropriate notions of post hoc explanations. Something that is akin to probability of sufficiency/ necessity etc (causal notions) are truly reliable explanations. Authors claim reliability but solve stability problems in prior post-hoc procedures.

Authors' contention: This dissenting opinion dismisses a whole  line of prior work on (popular) post-hoc explanations and we have sought to solve issue of these prior explanation methods.

The conflict at the heart seems to be : Are explanations provided by LIME, SHAP etc.. valid/reliable explanations in the first place. Solving stability issues of these seem to not address the core problem.

 My position is this: Probability of sufficiency/necessity are counterfactual notions that in general require knowledge about causal generative models behind the data. Sometimes, stronger assumptions like monotonicty etc.. (pls see - https://ftp.cs.ucla.edu/pub/stat_ser/r271-A.pdf) are required if we are to make conclusions from data alone. Several recent works are exploring, generalizing  and finding novel sufficient conditions for estimating these. But still they require some side information.

LIME, SHAP are perturbation based techniques based on the data manifold. While I agree with the dissenting opinion that they are not causally grounded (and hence not reliable in some absolute sense), in practice sometimes exploring the data manifold (in the factual sense) can provide explanations that are useful in practice. This is attested by the popularity of these methods and them being used in many places at this point. Another way to look at this is - given the popularity of these perturbation based notions on the data manifold, one would also want to see this line of enquiry mature given their easy to use data driven nature. I would not dismiss this line of enquiry completely.

Given the very favorable reviews from others, I am inclined to accept this paper. Authors may want to explicitly note that their method solves stability issues in a certain class of post hoc explanation methods and does not deal with explanation methods based on counterfactual notions.



**Consistency Experiment:**

NeurIPS has a long history of experimentation. In 2014, NeurIPS ran an experiment in which 10% of submissions were reviewed by two independent committees to quantify the randomness in the review process. This year, we repeated a variant of this experiment to see how the quality of the review process has changed over time.  This paper was part of the experiment and was therefore assigned to two committees (consisting of reviewers, an Area Chair, and a Senior Area Chair) that reached independent decisions.  If both committees made the same recommendation, this recommendation was followed. If a single committee recommended acceptance, the paper was accepted (with the exception of a few cases in which the other committee identified what we considered a fatal flaw, e.g., an error in a key result).

Both committees reached the same decision: **Accept (Poster)**

The other committee assigned to the paper recommended **Accept (Poster)**.  You can find the other set of reviews, along with any follow up discussion with the authors here:
https://openreview.net/forum?id=YUEFlzlG_0c